# Self-Paced Pairwise Representation Learning for Semi-Supervised Text Classification

## ABSTRACT

Text classification is one vital tool assisting web content mining. Modern deep learning approaches heavily rely on ample annotated data, which often comes at a considerable cost. Semi-supervised text classification (SSTC) offers an approach to alleviate the burden of annotation costs by harnessing the power of effective classifiers trained on a limited number of labeled texts alongside a vast pool of unlabeled texts. While existing SSTC methods have shown effectiveness by training a classifier on labeled texts and boosting the model with pseudo-labeled data derived from unlabeled texts, potential unsolved challenges are the overfitting problem caused by the limited availability of labeled data during training and the mislabeling problem stemming from an unreliable pseudo-labeling process. To address these issues, this paper proposes a **S**elf-**P**aced **P**air**W**ise representation learning (SPPW) model. Concretely, SPPW alleviates the overfitting problem by replacing the overfitting-prone learning of a parameterized classifier with representation learning in a pair-wise manner. Besides, our findings highlight the potential of utilizing text hardness as a complementary criterion to filter out unreliable texts upon existing confidence-based methods. With this insight, we propose a novel self-paced text filtering method that effectively integrates both label confidence and text hardness to reduce mislabeled texts synergistically. Extensive experiments on 3 benchmark SSTC datasets show that SPPW outperforms baselines and is effective in mitigating overfitting and mislabeling problems.

## CCS CONCEPTS

• **Computing methodologies → Natural language processing**.

## KEYWORDS

Text Classification, Semi-supervised Learning, Self-paced Learning

**ACM Reference Format:**
Anonymous Author(s). 2018. Self-Paced Pairwise Representation Learning for Semi-Supervised Text Classification. In *Proceedings of Make sure to enter the correct conference title from your rights confirmation emai (Conference acronym 'XX)*. ACM, New York, NY, USA, 10 pages. https://doi.org/XXXXXXX.XXXXXXX

## 1 INTRODUCTION

Text data constitutes a crucial element of web content. The mining of web text, including tasks such as clustering and classification,

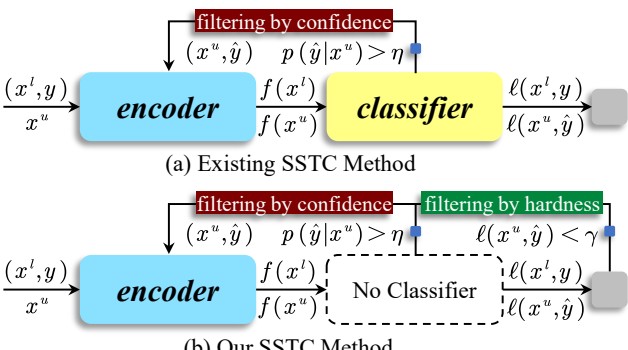

**Figure 1: Illustration of existing SSTC models (the upper part of the figure) and our SPPW model (the lower part of the figure).** $(x^l, y)$ **and** $x^u$ **represent the input labeled and unlabeled texts, respectively.** $f(\cdot)$ **denotes the representation of the input text.** $\hat{y}$ **is the pseudo-label for** $x^u$. $p(\hat{y}|x^u)$ **and** $\ell(x^u, \hat{y})$ **respectively denote the confidence and loss.** $\eta$ **and** $\gamma$ **are thresholds for filtering the pseudo-labeled texts.**

holds great significance for web applications. Modern deep learning approaches applied in text classification often require sufficient labeled data. However, collecting plenty of annotated text data is expensive in some real-world scenarios. Semi-supervised learning [1] that only requires a few labeled examples attached with many unlabeled examples can significantly reduce the reliance on laborious annotation. Semi-supervised text classification (SSTC) has recently been extensively studied [8, 9, 22, 32, 41]. To improve the SSTC results, existing works attempt to learn robust models by consistency training with the assistance of adversarial examples [29] or data augmentations [5, 39], or fine-tuning the model with the labeled texts under the regularization of unlabeled texts [10, 14]. To maximize the use of unlabeled texts, other works explore assigning pseudo labels for unlabeled texts and using them as additional training data to boost the model [5, 13, 20, 21, 27, 37, 39].

Despite their success, existing SSTC models still need to overcome two challenges that require further exploration: the *overfitting problem* caused by training the classifier using a few labeled texts and the *mislabeling problem* caused by assigning incorrect labels using the unreliable classifier as the pseudo-label model. The overfitting problem is common in SSTC because the training of the classifier relies on the few labeled texts that lead to a biased model. Furthermore, as shown in Figure 1 (a), since the existing SSTC models typically treat the classifier as the pseudo-label estimator for unlabeled texts, an overfitted classifier may receive inadequate accuracy and assign more error labels. For this reason, it is essential to guarantee the accuracy of the initial classifier, which governs the

label reliability assigned to unlabeled texts. Thus, we need to overcome the overfitting problem and provide a more reliable classifier for pseudo-labeling. Despite the partial mitigation of the overfitting problem through various regularization techniques in existing works, e.g., adversarial examples [29] or data augmentations [39], a promising solution lies in the exploration of alternative techniques that are not reliant on parameterized classifiers.

The mislabeling problem restricts the promotion gains from pseudo-labeled texts. Existing models leverage several useful strategies based on label confidence (as shown in Figure 1 (a)) to reduce unreliable labels assigned for unlabeled texts. For example, UDA [39] and MixText [5] utilize the threshold of the confidence score to filter the low-confidence examples and sharpen the predictions for consistency training. SALNet [21] leverages the high-confidence classification and lexicon predictions as pseudo-labels. Although these methods are effective, they only judge the validity of the pseudo-labeled texts based on the predicted label confidence of the classifiers. In this study, we uncover a novel insight unexplored in existing SSTC approaches: the potential of incorporating the hardness of texts as supplementary information of label confidence to improve the reliability of pseudo-labels.

Concretely, we propose a Self-Paced Pairwise representation learning (SPPW) model to address the problems mentioned above in SSTC. As shown in Figure 1 (b), instead of using regularization techniques, SPPW deals with the overfitting problem by converting the learning of the classification model (an encoder followed by a classifier) to only learning the representations (i.e., the encoder). We introduce a pairwise representation learning module to train the encoder to produce discriminative representations and infer the labels by aligning text representations with corresponding prototypes. This training strategy reduces learnable parameters and avoids directly learning the classifier tending to overfit with a few labeled texts, thus alleviating the overfitting problem.

Inspired by the self-paced learning technique [19] that gradually incorporates easier to harder samples into training, we propose a confidence-aware self-paced text filtering approach to deal with the mislabeling problem in SSTC. As shown in the lower part of Figure 1, we combine label confidence and text hardness to make a comprehensive decision on whether a pseudo-labeled text should be taken into training. The motivation to leverage text hardness as a reliability indicator is that the mislabeled texts tend to produce higher losses like hard examples (as analyzed in Figure 4 (b)). Thus, the mislabeled texts can be filtered by their hardness. The label confidence and text hardness serve as complementaries to filtering unreliable texts synergistically. Their interplay encourages self-paced learning to exclude more unreliable pseudo-labeled texts from training, thus mitigating the mislabeling problem in SSTC.

In a nutshell, our work makes the following contributions. (1) We propose a pairwise representation learning approach to avoid training the overfitting-prone classifier, which significantly alleviates the overfitting problem in SSTC. (2) We integrate label confidence and text hardness in self-paced learning to comprehensively filter unreliable texts, effectively mitigating the mislabeling problem in existing SSTC methods. (3) We conduct experiments on three datasets and empirically show that SPPW outperforms baselines on AGNews and DBPedia. The experiment analysis suggests that SPPW significantly mitigates the overfitting and mislabeling problems.

## 2 RELATED WORKS

Semi-supervised learning has become an emerging trend in text classification. One branch of works exploits regularization techniques or consistency training in SSTC that may alleviate the overfitting problem [14, 22, 24, 29, 40]. For example, UDA [39] substitutes noising operations and then optimizes the semi-supervised text classification model with consistency training. VAMPIRE [14] trains a variational auto-encoder with the unlabeled texts and utilizes it as a regularizer during training on labeled texts. Other works investigate data augmentations to compensate for the scarcity of labeled data [5, 39]. For example, MixText [5] proposes a new data augmentation method based on Mixup and includes the data augmentations and label sharpness in semi-supervised training. Another branch of research looks into assigning pseudo labels to unlabeled texts and utilizing them as additional training data [4, 13, 20–23, 27, 37, 38, 40, 43]. For example, SALNet [21] constructs lexicons based on attention weights and leverages the lexicons to improve pseudo-labeling and bootstrap the semi-supervised training. This study also builds upon the pseudo-labeling approach and aims to address the overfitting and mislabeling problems.

Learning text representations has been intensively studied in the Natural Language Processing community. Early works in sentence representation are mainly motivated by the idea of embedding [28], which learns representations based on the co-occurrence of n-gram words. The learned sentence representations are utilized to predict surrounding sentences [15, 18]. Recent studies have harnessed the power of pre-trained language models, such as BERT [10], to enhance text representation learning, showcasing their effectiveness and subsequent integration into numerous downstream applications. To further improve the expressive power, contrastive learning is leveraged for sentence representation learning [6, 35, 42, 44], which forces the representations of matched instances to be closer and unmatched instances to be distant. A similar idea has been adopted in pairwise learning method [2, 7], which attempts to learn discriminative representations in a pairwise manner. In this paper, we apply the pairwise learning paradigm to the mini-batch training of the text representations in SSTC. Unlike contrastive learning, our method does not rely on data augmentations and is designed to replace the classifier for mitigating overfitting.

Another direction related to our methods is the self-paced learning (SPL) approach. This learning paradigm is motivated by the human-learning process that gradually incorporates easier to harder samples during training [19]. The conventional self-paced learning model takes both the difficulty and diversity of training examples to rank the instances in self-paced learning [17]. The recent work investigates the impact of closely-coupled classes on adversarial attacks and develops a self-paced reweighting strategy in adversarial training [16]. Another work proposes a margin-preserving contrastive learning framework that utilizes self-paced learning for domain adaptation [25]. These methods use the hardness (or loss) of examples as critical evidence to determine the training order. In this study, we treat self-paced learning as a weighting technique to assess the reliability of pseudo labels, allowing us to filter mislabeled texts with improved accuracy and effectiveness. The concept of incorporating label confidence into self-paced learning holds significant potential for inspiring applications in various domains.

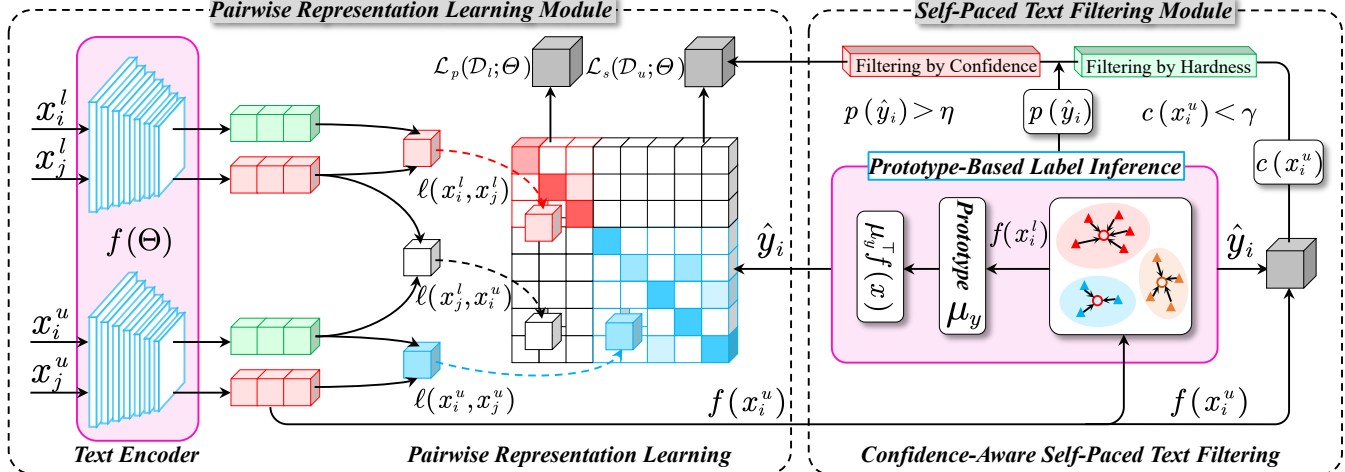

**Figure 2: The structure of our SPPW model.**

## 3 PROBLEM FORMULATION

Semi-supervised text classification (SSTC) attempts to learn effective text classification models from a training set that consists of a few labeled texts and plenty of unlabeled texts. Formally, let $\mathcal{Y}$ be the set of classes of interest, in which each class $y \in \mathcal{Y}$ denotes a specific text category from the class set. During training, we are given a small set of labeled-texts $\mathcal{D}_l = \{(x_1^l, y_1), (x_2^l, y_2), \cdots, (x_{n_l}^l, y_{n_l})\}$, where each class contains $K$ labeled texts, and a large set of unlabeled-texts $\mathcal{D}_u = \{x_1^u, x_2^u, \cdots, x_{n_u}^u\}$, where $(x_i^l, y_i)$ represent the $i^{\text{th}}$ labeled text and its label, $x_j^u$ denote the $j^{\text{th}}$ unlabeled text, $n_l$ and $n_u$ denote the number of labeled and unlabeled texts, respectively. In SSTC, the models need to train a classifier as robustly as possible with the few labeled texts and try to assist in training with unlabeled texts to the utmost to improve the model's performance.

## 4 METHODOLOGY

The structure of our SPPW model is shown in Figure 2, which consists of a pairwise representation learning module that combines a text encoder with the pairwise representation learning approach and a self-paced text filtering module that comprehensively takes the confidence of labels and the hardness of texts into account.

### 4.1 Pairwise Representation Learning Module

Our solution to the overfitting problem is reducing parameters by replacing the learning of the overfitting-prone classifiers with only learning the representations of texts. To that end, we propose a pairwise representation learning approach that respectively converges text representations of the same class and differentiates text representations from different classes in a pairwise way. To predict labels without a classifier, we introduce class prototypes and infer the labels of texts by aligning their representations with prototypes.

*4.1.1 Text Encoder.* Recent works introduce the pre-trained language models, such as BERT [10], as the encoder to learn text representations [11] and achieve remarkable performance. Existing

works in SSTC [5, 22, 39] utilize BERT as their text classification models for label prediction or pseudo-label estimation. Following these works, we also utilize BERT as our text encoder. Unlike existing SSTC models, we omit the classification component of BERT and only keep the encoding component as our text encoder. Specifically, given a text $x_i$ (either labeled or unlabeled), we get its BERT embedding $x_i \in \mathbb{R}^{d_x}$ by the following BERT encoding function

$$x_i = \text{BERT}(x_i, \theta), \tag{1}$$

where $\theta$ denotes the parameters of the BERT encoder. The text representation $z_i \in \mathbb{R}^{d_z}$ of $x_i$ is then obtained through a projection followed by a tanh activation function

$$z_i = \tanh(\mathbf{W}x_i + b), \tag{2}$$

where $\mathbf{W} \in \mathbb{R}^{d_z \times d_x}$ and $b$ respectively denote the parameters of the projection matrix and bias. For later use, we denote the text encoding process as a function $f(\cdot)$, i.e., $z_i = f(x_i)$ and represent its parameter set $\{\theta, \mathbf{W}, b\}$ as $\Theta$.

*4.1.2 Pairwise Representation Learning Approach.* Learning discriminative text representations in the supervised setting has been widely studied. The general idea is to pull closer the text representations with the same label and push away the representations with different labels. Inspired by the pairwise learning method used in [2, 7], we propose to apply the pairwise learning paradigm to the mini-batch training of the text representations.

Specifically, let $\mathcal{B}$ represent a batch of text examples drawn from the labeled or unlabeled text set. Then, for each text pair $x_i$ and $x_j$ in $\mathcal{B}$, the pairwise loss $\ell(x_i, x_j)$ is

$$\ell(x_i, x_j) = -\mathbb{I}_{y_i = y_j} \log[\sigma(f(x_i)^\top f(x_j))] \\ -(1 - \mathbb{I}_{y_i = y_j})\log[1 - \sigma(f(x_i)^\top f(x_j))], \tag{3}$$

where $\sigma$ is the logistic function. $y_i$ and $y_j$ respectively denote the labels (or pseudo labels) of the $i^{\text{th}}$ and $j^{\text{th}}$ example in batch $\mathcal{B}$. $\mathbb{I}$ indicates whether two texts have the same label. Namely, if $y_i = y_j$, we have $\mathbb{I}_{y_i = y_j} = 1$, else if $y_i \neq y_j$, then we get $\mathbb{I}_{y_i = y_j} = 0$.

Suppose we use $\mathcal{D}$ to denote the training dataset (either from $\mathcal{D}_l$ or $\mathcal{D}_u$). To perform pairwise representation learning, we can minimize the following objective

$$\mathcal{L}_p(\mathcal{D}; \Theta) = \sum_{\mathcal{B} \subseteq \mathcal{D}} \sum_{(x_i, x_j) \in \mathcal{B}} \ell(x_i, x_j) \tag{4}$$

The aforementioned pairwise loss fosters the generation of discriminative text representations while mitigating overfitting issues by excluding learnable classifiers. As a result, it strengthens the model and offers a more reliable pseudo-label estimator for assigning labels to unlabeled texts. The pairwise loss will be first used to pre-train the representation learning model with labeled texts and then fine-tune the model with labeled and pseudo-labeled texts.

*4.1.3 Prototype-Based Label Inference.* Since the representation learning model is trained without classifiers, we are unable to make predictions based on any trained classifier. Hence, we must rely solely on the learned text representations to infer labels. Fortunately, the prototypes can be utilized for label inference even in the absence of a classifier. This approach has been widely used in few-shot learning [6, 33, 36]. Specifically, to obtain the prototype for each class $y \in \mathcal{Y}$, we randomly sample $k$ examples from the $K$ labeled texts of each class in $\mathcal{D}_l$. Let $\mathcal{P} = \{(x_1^l, y_1), (x_2^l, y_2), \cdots, (x_{k \times |\mathcal{Y}|}^l, y_{k \times |\mathcal{Y}|})\}$ denote the sampled $k \times |\mathcal{Y}|$ labeled texts, then we can compute the prototype $\mu_y$ for class $y$ by the following formulation

$$\mu_y = \frac{1}{k} \sum_{(x_i^l, y_i^l) \in \mathcal{P}} \mathbb{I}_{y_i = y} f(x_i^l) \tag{5}$$

Let $p(y|x)$ denote the probability (or confidence) of $x$ belonging to label $y$, which is defined as follows

$$p(y|x) = \frac{\exp(\mu_y^\top f(x))}{\sum_{y' \in \mathcal{Y}} \exp(\mu_{y'}^\top f(x))} \tag{6}$$

Then, the inferred label for input text $x$, which is denoted as $\hat{y}$, can be found by the following argmax operation

$$\hat{y} = \arg \max_{y \in \mathcal{Y}} p(y|x) \tag{7}$$

The above label inference function can be used to estimate the pseudo-label of an unlabeled text during training or predict the label of a test example during evaluation.

## 4.2 Self-Paced Text Filtering Module

After training the model on the labeled texts in $\mathcal{D}_l$, we may assign pseudo-labels for the unlabeled texts in $\mathcal{D}_u$ and include these texts to further fine-tune the model. To mitigate the mislabeling problem when pseudo-labeling unlabeled texts during fine-tuning, we propose a confidence-aware self-paced learning approach to filter out unreliable texts, excluding mislabeled texts from training by considering both label confidence and text hardness.

*4.2.1 Prototype Calibration.* During the fine-tuning stage, if the labeled texts used to compute the prototypes are excluded from training, the learned representations may shift from the prototypes and degrade the accuracy of label inference. To fix this problem,

we introduce the following prototype calibration loss for each unlabeled text $x_i^u \in \mathcal{D}_u$ defined as

$$c(x_i^u) = -\log \frac{\exp(\mu_{\hat{y}_i}^\top f(x_i^u))}{\sum_{y \in \mathcal{Y}} \exp(\mu_y^\top f(x_i^u))} \tag{8}$$

The calibration loss can be trained in conjunction with the pairwise representation learning loss to ensure alignment between text representations and their corresponding prototypes, thereby preventing undesired shifts in the text representations.

*4.2.2 Text Filtering with Label Confidence.* Existing works filter unreliable pseudo-labeled texts by setting label confidence criteria based on the assumption that reliable pseudo-labeled texts are often associated with high label confidence [5, 21, 27, 37, 39]. In this paper, we also utilize label confidence as an indicator to filter mislabeled texts pseudo-labeling from the unlabeled text set.

Specifically, we introduce two binary indicators $\alpha_{ij}, \beta_i \in \{0, 1\}$ attached to the pairwise loss $\ell(x_i^u, x_j^u)$ and the prototype calibration loss $c(x_i^u)$, respectively, each determining whether the loss will be maintained for updating the model parameters or discarded. We define the resulting objective as

$$\mathcal{L}_c(\mathcal{D}_u; \Theta) = \sum_{\mathcal{B} \subseteq \mathcal{D}_u} \sum_{(x_i^u, x_j^u) \in \mathcal{B}} \alpha_{ij} \ell(x_i^u, x_j^u)$$
$$+ \sum_{\mathcal{B} \subseteq \mathcal{D}_u} \sum_{x_i^u \in \mathcal{B}} \beta_i c(x_i^u) \tag{9}$$

The values of $\alpha_{ij}$ and $\beta_i$ are determined by the label confidence of the pseudo-labeled text as follows

$$\alpha_{ij} = \begin{cases} 1 & p(\hat{y}_i), p(\hat{y}_j) > \eta, \\ 0 & otherwise. \end{cases} \quad \beta_i = \begin{cases} 1 & p(\hat{y}_i) > \eta, \\ 0 & otherwise. \end{cases} \tag{10}$$

where the probabilities $p(\hat{y}_i)$ and $p(\hat{y}_j)$ denote the label confidence $p(\hat{y}_i|x_i^u)$ and $p(\hat{y}_j|x_j^u)$, respectively. And the parameter $\eta$ is a threshold used to control the values of $\alpha_{ij}$ and $\beta_i$, which is treated as a hyper-parameter during training.

*4.2.3 Confidence-Aware Self-Paced Learning Approach.* The concept of self-paced learning suggests that model generalization can be enhanced by initially training the model with easier examples and progressively introducing harder ones [17]. Building upon this notion, we expect to employ self-paced learning to filter out unreliable pseudo-labeled texts based on the hardness of examples. Our findings from the empirical studies (see Figure 4 (b)) suggest that mislabeled texts tend to output large losses like hard examples, allowing us to filter them based on their losses (or hardness). Thus, we design the following confidence-aware self-paced learning loss

$$\mathcal{L}_s(\mathcal{D}_u; \Theta) = \sum_{\mathcal{B} \subseteq \mathcal{D}_u} \sum_{(x_i^u, x_j^u) \in \mathcal{B}} \alpha_{ij} \ell(x_i^u, x_j^u)$$
$$+ \sum_{\mathcal{B} \subseteq \mathcal{D}_u} \sum_{x_i^u \in \mathcal{B}} \beta_i [w_i c(x_i^u) - \gamma w_i], \tag{11}$$

where the calibration loss $c(x_i^u)$ is used as the indicator to measure the hardness of $x_i^u$, because it reflects how much $x_i^u$ matches with its pseudo-label, i.e., a higher calibration loss implies that the example is a harder example. The parameter $w_i \in \{0, 1\}$ is a binary weight on the loss $c(x_i^u)$, i.e., when $w_i = 0$, the loss $c(x_i^u)$ will be filtered

and will not be used to update the model parameters. The hyper-parameter $\gamma$ is a parameter to control the learning pace. According to [17], $w_i$ has a global optimum with fixed $\Theta$ as

$$w_i = \begin{cases} 1 & c(x_i^u) < \gamma, \\ 0 & otherwise. \end{cases} \quad (12)$$

Here $\gamma$ could be viewed as a threshold to filter the pseudo-labeled texts according to the hardness of the texts.

We additionally design a confidence-aware value adaptation strategy to adjust the learning pace $\gamma$ during the fine-tuning stage. Specifically, we call every $m$ SGD steps a training episode and will update $\gamma$ every training episode. In the current training episode, we compute a temporal loss $\tau$ by weighted sum losses in the training episode, i.e., $\tau = \sum_i p(\hat{y}_i)c(x_i^u)$. $\gamma$ is then updated by $\gamma = \gamma' \tau / \tau'$, where $\gamma'$ and $\tau'$ are the learning pace and temporal loss of the last training episode. Note that the label confidence $p(\hat{y}_i)$ will be leveraged to weigh the losses when updating parameter $\gamma$.

*4.2.4 The SSTC Training Procedure.* Our SSTC model combines the pairwise representation learning module and the self-paced text filtering approach. We can adopt an iterative optimization process like self-training [12, 13, 20, 21] to train our model by the following pre-training and fine-tuning stages:

(1) Pre-training with labeled texts: Pre-train the pairwise representation learning module using the labeled text dataset $\mathcal{D}_l$ by minimizing the summed pairwise loss $\min_\Theta \mathcal{L}_p(\mathcal{D}_l; \Theta)$ and use it for prediction. For later use, we name the model trained in this stage **PW**. This model is used as a baseline.

(2) Fine-tuning with labeled and unlabeled texts: Estimate pseudo-labels for the unlabeled texts in $\mathcal{D}_u$ via label inference in Equation (7) with current parameter $\Theta$ of PW, then fine-tune the model by minimizing the loss $\mathcal{L}_s(\mathcal{D}_u; \Theta)$ with pseudo-labeled texts filtered by label confidence and text hardness. This model is our ultimate model and we name it **SPPW**.

## 5 EXPERIMENT

### 5.1 Datasets and Experiment Setting

Following the previous works in semi-supervised text classification [5, 21, 22, 39], we evaluate SPPW on three datasets: AGNews, DBPedia and Yahoo.

**AGNews** [31] is a subdataset of AG news created by compiling the titles and descriptions of articles. It contains 127600 texts examples from the 4 categories, including *World*, *Sports*, *Business* and *Sci/Tech*.

**DBPedia** [26] is a query understanding dataset extracted from Wikipedia. This dataset contains 630000 texts from 14 classes for text classification, including *Company*, *Educational Institution*, etc.

**Yahoo** [3] is a question classification dataset. The question/answer pairs are extracted from the Yahoo! Answers website with 10 top-level categories, which contains 1460000 texts from 10 classes, including *Society & Culture*, *Health*, *Education & Reference*, etc.

We use the available datasets in MixText [5] and split training sets to $K = 10$, 50 and 200 labeled texts and 5000 unlabeled texts for each class. We keep the unlabeled texts, validation, and test sets the same as MixText. Following the previous works, we use FairSeq[1] to get back-translated texts as data augmentations for unlabeled

[1]https://github.com/pytorch/fairseq

**Table 1: Statistics of the datasets. # labeled denote labeled texts for each class, and #unl., #val and #test denote total examples for unlabeled, validation and test set, respectively.**

| Dataset | #labeled | #unl. | #val | #test | $|\mathcal{Y}|$ |
|---------|----------|-------|------|-------|-----|
| AGNews | 10/50/200 | 20000 | 8000 | 7600 | 4 |
| DBPedia | 10/50/200 | 70000 | 28000 | 70000 | 14 |
| Yahoo | 10/50/200 | 50000 | 50000 | 60000 | 10 |

texts. We adopt Accuracy (Acc) and F1 as the evaluation metrics. The statistics of the datasets are shown in Table 1.

### 5.2 Implementation and Baseline Models

*5.2.1 Implementation.* Our model is implemented with PyTorch and is released anonymously for reproduction[2]. All hyper-parameters are selected by grid search on the validation set. The dimension of word embedding $d_x$ is 768. The hidden size $d_z$ of SPPW is set as 128 on AGNews and DBPedia, and 512 on Yahoo. The training batch size is set to 8. During pre-training PW, we adopt early stopping based on the performance of the validation set. To obtain prototypes, we set $k = 10$ for $K = 10$ setting and $k = 20$ on DBPedia, $k = 50$ on AGNews, Yahoo for other settings. During the fine-tuning stage, we set $\eta = 0.95, 0.7, 0.9$ respectively on AGNews, DBPedia and Yahoo for the self-paced text filtering approach. On AGNews and DBPedia, $\gamma$ is initialized with 0.3, and we set $m = 6$ for a training episode. On Yahoo, the training episode is set to $m = 7$, and $\gamma$ is initialized with 1. The learning rate is set to $1e-5$ during pre-training and $1e^{-7}$ for fine-tuning on Yahoo, and $1e^{-3}$ during pre-training and $1e^{-5}$ for fine-tuning on other datasets. All experiments are conducted on an NVIDIA A100-PCIE GPU with 40GB memory.

*5.2.2 Baseline Models.* We evaluate the models on the three benchmarks and compare them with the following baselines:

**BERT** [10] utilizes the BERT-based-uncased model for text classification without using the unlabeled texts and data augmentations for additional training.

**UDA** [39] substitutes noising operations with data augmentations and then optimizes the semi-supervised text classification model with consistency training.

**MixText** [5] proposes a new data augmentation method based on Mixup and includes the data augmentations and label sharpness in consistency training.

**SALNet** [21] constructs lexicons based on attention weights and leverages the lexicons to improve pseudo-labeling and bootstrap the semi-supervised training.

To make fair comparisons with the baselines, we use the released code in their original papers to run experiments with the split data. For BERT, UDA, MixText, PW and SPPW, we utilize the same BERT encoder and data augmentations as MixText. We run each model 5 times for all experimental settings and report the mean and standard deviation. To showcase the superior efficacy of our model, we further conduct a comprehensive comparison with the reported results of recent SSTC models, including FixMatch [34], UST [30], FLiText [24], $S^2$TC-BDD [22], DPS [23], SAT [4] and CEST [38].

[2]Anonymous link for code and data: https://file.io/xJeAgDGjnusb

**Table 2: The semi-supervised text classification results on AGNews, Yahoo and DBPedia. [†] Note that the reproduced results are generally consistent with the reported results in the original paper with a similar number of labeled texts.**

| Method | AGNews | | | | | | Yahoo | | | | | | DBPedia | | | | | |
|---|---|---|---|---|---|---|---|---|---|---|---|---|---|---|---|---|---|---|
| | 10 | | 50 | | 200 | | 10 | | 50 | | 200 | | 10 | | 50 | | 200 | |
| | Acc | F1 | Acc | F1 | Acc | F1 | Acc | F1 | Acc | F1 | Acc | F1 | Acc | F1 | Acc | F1 | Acc | F1 |
| BERT | 72.65 | 70.14 | 85.31 | 85.32 | 87.98 | 87.84 | 56.11 | 55.80 | 65.19 | 65.08 | 68.78 | 68.63 | 96.08 | 95.86 | 98.35 | 98.34 | 98.82 | 98.81 |
| | ±1.77 | ±2.01 | ±0.05 | ±0.10 | ±0.20 | ±0.40 | ±1.04 | ±1.15 | ±0.53 | ±0.66 | ±0.34 | ±0.24 | ±0.18 | ±0.29 | ±0.12 | ±0.12 | ±0.03 | ±0.04 |
| UDA | 84.15 | 84.10 | 88.02 | 88.01 | 88.92 | 88.91 | 62.83 | 59.91 | 68.42 | 67.35 | 70.52 | 70.05 | 98.38 | 98.37 | 98.85 | 98.84 | 98.92 | 98.92 |
| | ±1.10 | ±1.05 | ±0.31 | ±0.32 | ±0.10 | ±0.12 | ±0.88 | ±1.36 | ±0.57 | ±0.68 | ±0.28 | ±0.14 | ±0.21 | ±0.21 | ±0.04 | ±0.04 | ±0.03 | ±0.03 |
| MixText | 86.57 | 86.28 | 87.24 | 87.17 | 88.65 | 88.54 | **65.40** | **64.25** | 68.69 | 67.92 | 70.49 | 69.97 | 97.67 | 97.67 | 98.46 | 98.46 | 98.83 | 98.82 |
| | ±0.56 | ±0.61 | ±0.42 | ±0.44 | ±0.11 | ±0.05 | ±1.72 | ±1.46 | ±0.44 | ±0.45 | ±0.29 | ±0.22 | ±0.09 | ±0.10 | ±0.03 | ±0.03 | ±0.06 | ±0.07 |
| SALNet[†] | 77.61 | 77.61 | 86.17 | 86.21 | 88.25 | 88.23 | 52.43 | 52.30 | 53.65 | 53.45 | 59.08 | 59.02 | 95.39 | 95.39 | 97.08 | 97.08 | 98.66 | 98.65 |
| | ±3.17 | ±3.17 | ±0.32 | ±0.36 | ±0.15 | ±0.14 | ±0.27 | ±0.16 | ±0.95 | ±1.02 | ±0.76 | ±0.52 | ±0.17 | ±0.15 | ±0.25 | ±0.25 | ±0.04 | ±0.05 |
| **PW** | 81.13 | 81.13 | 86.38 | 86.33 | 87.89 | 87.86 | 62.89 | 61.03 | 66.67 | 66.21 | 69.07 | 68.71 | 96.97 | 96.96 | 98.42 | 98.42 | 98.76 | 98.75 |
| | ±2.23 | ±2.24 | ±0.48 | ±0.53 | ±0.14 | ±0.14 | ±1.93 | ±3.03 | ±0.46 | ±0.57 | ±0.26 | ±0.18 | ±0.23 | ±0.23 | ±0.08 | ±0.07 | ±0.03 | ±0.04 |
| **SPPW** | **88.59** | **88.54** | **89.13** | **89.10** | **89.38** | **89.36** | 64.86 | 63.76 | **68.80** | **67.95** | **71.12** | **70.39** | **98.43** | **98.43** | **98.88** | **98.88** | **98.93** | **98.93** |
| | ±0.44 | ±0.43 | ±0.14 | ±0.14 | ±0.11 | ±0.11 | ±1.48 | ±1.75 | ±0.13 | ±0.30 | ±0.09 | ±0.15 | ±0.25 | ±0.27 | ±0.07 | ±0.06 | ±0.03 | ±0.02 |

**Table 3: Additional results compared to reported results in recent works. The bold values denote the best results.**

| AGNews | | | | |
|---|---|---|---|---|
| Scale of K | Model | K | Acc | F1 |
| | **SPPW** | 10 | **88.59** | **88.54** |
| | FixMatch | 10 | 80.22 | 79.98 |
| $10 \leqslant K < 50$ | SAT | 10 | 86.38 | 86.29 |
| | UST | 30 | 86.90 | - |
| | CEST | 30 | 87.05 | - |
| | **SPPW** | 50 | **89.13** | **89.10** |
| $50 \leqslant K < 200$ | S$^2$TC-BDD | 100 | - | 87.20 |
| | DPS | 100 | - | 88.70 |
| $K \geqslant 200$ | **SPPW** | 200 | **89.38** | **89.36** |
| Yahoo | | | | |
| | **SPPW** | 10 | **64.86** | **63.76** |
| $10 \leqslant K < 50$ | FixMatch | 10 | 60.17 | 59.86 |
| | SAT | 20 | 61.51 | 61.09 |
| | **SPPW** | 50 | **68.80** | **67.95** |
| $50 \leqslant K < 200$ | S$^2$TC-BDD | 100 | - | 61.80 |
| | DPS | 100 | - | 63.20 |
| $K \geqslant 200$ | **SPPW** | 200 | **71.12** | **70.39** |
| | FLiText | 500 | 65.08 | - |
| DBPedia | | | | |
| | **SPPW** | 10 | 98.43 | **98.43** |
| $10 \leqslant K < 50$ | UST | 30 | 98.30 | - |
| | CEST | 30 | **98.61** | - |
| $50 \leqslant K < 200$ | **SPPW** | 50 | **98.88** | **98.88** |
| $K \geqslant 200$ | **SPPW** | 200 | **98.93** | **98.93** |

## 5.3 SSTC Evaluation Results

*5.3.1 Comparison with Baseline Models.* The semi-supervised text classification results of the compared models are shown in Table 2. It illustrates that SPPW outperforms baseline models in all settings except the $K = 10$ setting on Yahoo. This observation manifests that our SPPW model achieves improvements in SSTC, especially when the labeled data is rare, e.g., $K = 10$ setting on AGNews, suggesting that our pair-wise representation learning combined with self-paced text filtering is effective for SSTC. Between the models only trained with labeled texts, PW significantly outperforms the BERT model with a few labeled texts (e.g., about 7% and 2% gains on AGNews and Yahoo with 10 labeled texts per class), which demonstrates that mitigating the overfitting problem using PW helps improve the SSTC performance. Another observation is that SPPW achieves little gain or performs worse than MixText on Yahoo. The reason behind this may be attributed to the implementation of the pseudo-labeling framework in SPPW, as its effectiveness may be tied to the performance of the initial pseudo-labeler. Thus, SPPW achieves limited gain on Yahoo with a relatively poor performance of PW.

*5.3.2 Comparison with Reported Results of Recent Works.* To make a comprehensive evaluation, we compare SPPW with the reported results of the recent semi-supervised text classification model in Table 3. Acknowledging the potential discrepancies in data splitting and experimental settings across various studies, it becomes crucial to recognize the challenges of making a fair comparison with them. However, to ensure an equitable evaluation, we conduct our analysis under the constraint that our model utilizes an equal or smaller number of $K$ labeled texts. Namely, our model is *disadvantaged* in comparison because the results are significantly influenced by the quantity of labeled data. The results show that SPPW achieves the best results on almost all settings of all datasets, even with fewer labeled texts. The obtained results clearly demonstrate the effectiveness of SPPW in semi-supervised text classification, establishing it as a robust and powerful model in this domain.

**Table 4: The evaluation results of ablated models. We report the ablation study results on the AGNews dataset.**

| Model | $K = 10$ |
|---|---|
| PW | 81.13 |
| +hardness filtering component ($\gamma$) | 85.64 |
| +confidence filtering component ($\eta$) | 86.93 |
| +$\gamma$+$\eta$ | 87.22 |
| +$\gamma$+data augmentation | 87.82 |
| +$\eta$+data augmentation | 88.01 |
| +$\gamma$+$\eta$+data augmentation (SPPW) | **88.59** |

*5.3.3 Ablation Study.* To analyze the contributions of each component in our model, we make ablation studies on three components of SPPW: the data augmentations, the hardness filtering component with threshold $\gamma$ and the confidence filtering component with threshold $\eta$. As illustrated in Table 4, adding the hardness or confidence filtering component improves upon PW. This result suggests that both label confidence and text hardness may help filter unreliable pseudo-labeled texts and the designed text filtering approaches in our model indeed improve the SSTC performance. The model utilizing both the hardness filtering and confidence filtering components outperforms the models that only use hardness filtering or confidence filtering. This result demonstrates that the interplay of hardness and confidence in our self-paced filtering approach effectively excludes unreliable pseudo-labeled texts and boosts SSTC training. The data augmentations further improve our models and build SPPW as the new state-of-the-art on AGNews.

## 5.4 The Detailed Analysis of SPPW

*5.4.1 Effectiveness in Alleviating Overfitting.* To investigate the effectiveness of our pair-wise representation learning module in alleviating the overfitting problems, we report the training and validation records of the pure BERT model shared in most baselines and our PW model in different training epochs in Figure 3. As shown in Figure 3 (a), the training accuracy of BERT quickly increases to 1 at the early training stage (near epoch 5). Conversely, the training accuracy of PW increases relatively slower and reaches 1 at later training stage near epoch 50. In Figure 3 (b), the validation accuracy of BERT increases quickly and achieves the peak (about 0.7) at the early training stage near epoch 10. But the validation accuracy of PW slowly increases and achieves a much better peak value (about 0.8) at the latter training stage near epoch 60. The validation accuracy of PW continues increasing after 50 epochs because the training loss still decreases, although the training accuracy nearly stops increasing. The curves imply that BERT is prone to overfit on a few labeled texts, and PW significantly mitigate the overfitting problem by learning representations with pair-wise losses.

*5.4.2 Effectiveness in Text Filtering.* A precise text filtering approach holds paramount importance in SSTC as it ensures the effectiveness of leveraging pseudo-labeled texts to enhance model training. To study our self-paced approach, we analyse the text filtering process in training. Specifically, we respectively combine the PW model with the confidence filtering component $\eta$ and the

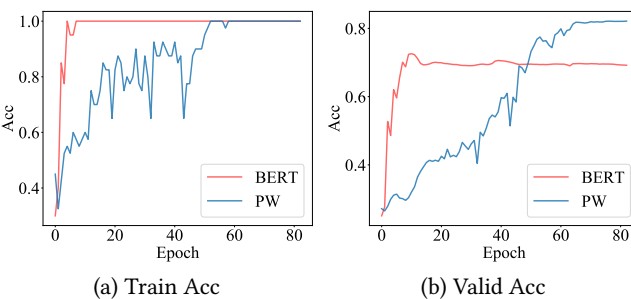

(a) Train Acc  (b) Valid Acc

**Figure 3: The curves of training-accuracy (a) and validation-accuracy (b) for BERT and PW on the AGNews dataset.**

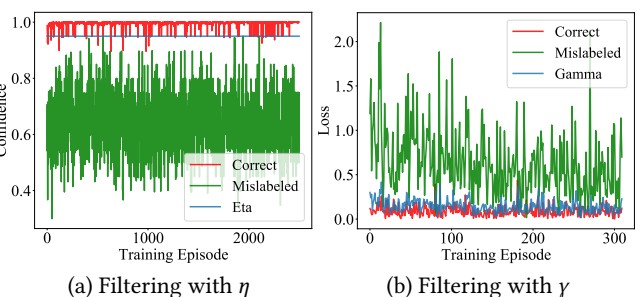

(a) Filtering with $\eta$  (b) Filtering with $\gamma$

**Figure 4: The text filtering processes on AGNews.**

hardness filtering component $\gamma$ and record the average label confidence and text hardness (or loss) over the correctly pseudo-labeled texts and mislabeled texts. The visualization results are illustrated in Figure 4, which shows that the mislabeled texts generally have lower confidence and higher hardness than the correctly labeled texts. Our hyper-parameter setting $\eta = 0.95$ and the adapted $\gamma$ generally separate the correctly labeled and mislabeled texts. These visualization results demonstrate that our self-paced text filtering approach that considers both label confidence and text hardness is effective in excluding unreliable pseudo-labeled texts.

*5.4.3 The Interplay Between $\eta$ and $\gamma$ Filtering.* To study the interplay between the confidence filtering component $\eta$ and the hardness filtering component $\gamma$, we report the number of texts that are kept for training in an epoch in Figure 5 (a) and the number of mislabeled texts in these kept examples in Figure 5 (b) after filtering by $\eta$, $\gamma$, or both $\eta$ and $\gamma$ (All). From Figure 5 (a), we observe that using the $\eta$ component to filtering texts will keep more texts in training than using the $\gamma$ component. When we combine $\eta$ and $\gamma$ components for text filtering, much fewer texts will be kept in training, i.e., more pseudo-labeled texts are filtered than using only $\eta$ or $\gamma$ component. From Figure 5 (b), we observe that when both $\eta$ and $\gamma$ components are not used (Total), around $3400 \sim 3600$ mislabeled texts will be included into the training. Using either $\eta$ or $\gamma$ for filtering reduces the mislabeled texts and using both $\eta$ and $\gamma$ for filtering (All) keeps the least mislabeled texts. These results suggest that combining the $\eta$ and $\gamma$ components for text filtering excludes more mislabeled texts from training than using either of them.

**Table 5: Case study on two selected text examples from the AGNews dataset.**

| id | Text | Golden Label | Pesudo Label | Loss | $\gamma$ | Confidence | $\eta$ |
|----|------|--------------|--------------|------|----------|------------|--------|
| 1 | SAN FRANCISCO - Omar Vizquel hopes to revitalize his career with a new team in a new league. The San Francisco Giants just hope the veteran shortstop has a few more good years in him. | Sports | Business | 0.119 | 0.240 | 0.5684 | 0.95 |
| 2 | Internet users at home are not nearly as safe online as they believe, according to a nationwide inspection by researchers. | Sci/Tech | World | 0.028 | 0.006 | 0.9680 | 0.95 |

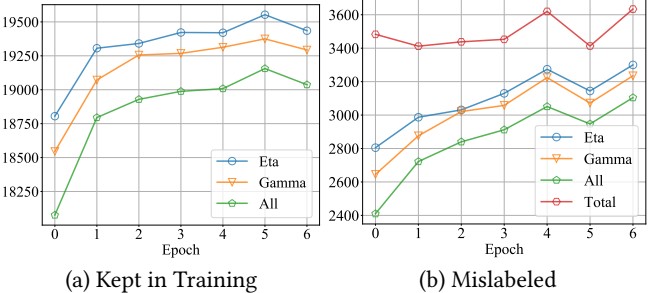

(a) Kept in Training  (b) Mislabeled

**Figure 5: The interplay between $\eta$ and $\gamma$ on AGNews.**

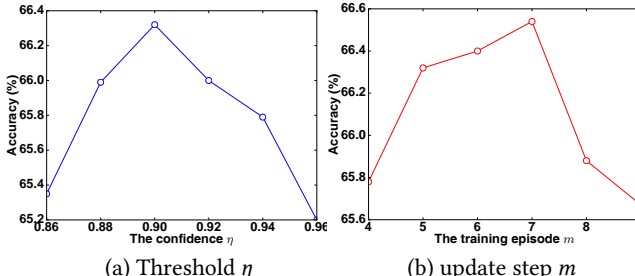

(a) Threshold $\eta$  (b) update step $m$

**Figure 6: Analysis of the confidence threshold $\eta$ and update step $m$ on AGNews.**

*5.4.4 Impact of Confidence Threshold $\eta$.* In the self-paced text filtering module, the hyper-parameter $\eta$ is treated as the threshold to identify unreliable pseudo-labeled texts, which is important for text filtering with label confidence. To study whether SPPW is sensitive to the confidence threshold $\eta$, we make an analysis on different settings of hyper-parameter $\eta$. Concretely, we first reduce the search range according to the confidence score on the validation set, then train SPPW with different $\eta$ within the search range. The evaluation results with various $\eta$ on AGNews are shown in Figure 6 (a). The performance change in the figure manifests that the model performance is sensitive to the confidence threshold $\eta$. Thus, $\eta$ needs to be searched on the validation set for different datasets. The results show that the best configuration of $\eta$ on AGNews is 0.90.

*5.4.5 Impact of Update Step $m$.* In our self-paced text filtering approach, we use the hyper-parameter $\gamma$ as the threshold to exclude unreliable texts based on their hardness. The configuration of $\gamma$ is not sensitive to the model performance because it is automatically updated based on the training status. Nevertheless, the number of SGD steps $m$ in each training episode can be sensitive as it governs the pace at which the learning progresses and updates. Too large or too small $m$ may result in sub-optimal updating of $\gamma$. To analyze how $m$ affects the text filtering process. We report the results of SPPW with different $m$ in Figure 6 (b). The results demonstrate that the update step $m$ indeed affects the training of SPPW and the optimal value of $m$ can be easily determined by grid search on the validation set. For example, the optimal $m$ on AGNews is 7.

*5.4.6 Case Study.* Text filtering with either $\eta$ or $\gamma$ component may sometimes fail. Nevertheless, when the two components are combines, these mislabeled texts may be correctly filtered. To verify

this, we study on two examples selected from AGNews in Table 5. In case 1, the text is pseudo-labeled as *Business*, but the golden label is *Sports*. This text output a loss of 0.119 and a confidence of 0.5684. If we only use the hardness filtering component, this mislabeled text will be taken into training because its loss $0.119 < \gamma = 0.240$. Nevertheless, if we incorporate the confidence filtering component, it will be filtered because its confidence $0.5684 < 0.95 = \eta$. Similarly, in case 2, if we only use the confidence filtering component, the mislabeled text will include in training because its confidence $0.9680 > 0.95 = \eta$. And if we also integrate the hardness filtering component, this mislabeled text will be filtered because its loss $0.028 > 0.006 = \gamma$. These cases suggest that considering label confidence and text hardness is necessary for SSTC and demonstrate that our self-paced text filtering approach is effective.

## 6 CONCLUSION

We introduce a self-paced pairwise representation learning (SPPW) model as a solution to address the challenges of overfitting and mislabeling in SSTC. SPPW mitigates the issue of overfitting by replacing the learnable classifiers with pairwise representation learning, while simultaneously reducing mislabeled texts through self-paced text filtering that considers both label confidence and text hardness. Empirical studies on three benchmarks show that SPPW outperforms baseline models and effectively mitigates the overfitting and mislabelling problems. Our pairwise representation learning method has the potential to be extended to other classification tasks characterized by limited labeled data, such as few-shot learning. The self-paced filtering method, which takes into account both the confidence and hardness of examples, offers an alternative for training machine learning models under unreliable data.

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

# A  APPENDIX A

## A.1  Additional Results and Analysis

*A.1.1  Training with Various Number of Unlabeled Texts.* To evaluate the models with different number of unlabeled texts, we compare the models training with 1000 / 3000 / 5000 / 7000 unlabeled texts for each class on AGNews. The evaluation results in Table 6 show that the performance of both Mixtext and SPPW increase when the unlabeled texts increase, but SPPW perform better than Mixtext when using fewer unlabeled texts for training. This fact demonstrate that our SPPW model can more efficiently learn from the few labeled data. Another observation is that when the unlabeled texts are more than 5000 for each class, the models only get little improvements. Thus, we fixed the unlabeled texts for each class as 5000 when evaluating the models, which is the same as in the Mixtext paper.

**Table 6: The evaluation results of using various number of unlabeled texts on the AGNews dataset.**

| Model | 1000 | 3000 | 5000 | 7000 |
|---|---|---|---|---|
| Mixtext | 82.61 | 84.33 | 86.80 | 87.21 |
| SPPW | 87.52 | 88.34 | 89.21 | 89.28 |

*A.1.2  Evaluation on Synthetic Imbalanced Datasets.* To examine whether the models work well on imbalanced unlabeled texts, we create synthetic datasets using from the AGNews dataset and use them to train Mixtext and SPPW. Specifically, for the 4 classes in AGNews, we random select 1000 / 2000 / 5000 / 8000 texts as imbalanced unlabeled texts. For simplicity, we use [1:2:5:8] to denote the ratio of unlabeled texts for the 4 classes, and we report the results in Table 7. The results show that Mixtext is sensitive to unbalanced data and SPPW is robust to unbalanced data. The main reason we guess is that our pair-wise representation learning is not sensitive to unbalanced data.

**Table 7: The evaluation results on synthetic imbalanced datasets constructed from the AGNews dataset.**

| Model | [5:1:2:8] | [8:5:2:1] | [2:8:5:1] | Average |
|---|---|---|---|---|
| Mixtext | 86.46 | 83.78 | 76.89 | 80.34 |
| SPPW | 87.58 | 87.84 | 87.13 | 87.52 |

**Table 8: The evaluation results of SPPW using different batch sizes on the AGNews dataset.**

| Batch Size | Performance (Accuracy) | Average |
|---|---|---|
| 4 | 86.78 / 88.32 / 88.67 | 88.59 |
| 8 | 88.47 / 88.72 / 88.20 | 88.46 |
| 16 | 89.21 / 88.22 / 88.33 | 88.59 |

*A.1.3  Impact of Batch Size.* Our pair-wise representation learning select texts pairs within a batch of data. Thus, the batch size may impact the training efficiency. To investigate how batch size impacts the training of SPPW, we run SPPW with batch size 4 / 8 / 16 three times and report the results in Table 8. The results show that the batch size has little impacts on the averaged performance of SPPW. It means that our method is not sensitive to batch size.

