# OpenReview forum: "Self-Paced Pairwise Representation Learning for Semi-Supervised Text Classification"
_ACM.org/TheWebConf/2024/Conference — TheWebConf24_

### Official Review · Reviewer_PHr4 · 2023-11-20

**Novelty:** 4
**Technical Quality:** 4

**Review:**

This paper addresses issues encountered in semi-supervised learning, such as overfitting and the thresholding approach for filtering low-confidence samples. The paper tackles an important problem commonly found in industrial setups. The problem and solution are written clearly in the paper.

**Questions:**

1. The author claims to handle overfitting by reducing the number of parameters (line 278). I am unsure about how the parameters are reduced. In typical BERT-based text classification solutions, we project the embedding corresponding to the [CLS] token into lower dimensions using an MLP. Here, for effective text representation, it's mentioned that projection into lower dimensions is done (line 326).

2. I am uncertain about the theoretical rationale behind this approach having an extra edge compared to a simple thresholding approach. It seems labels are determined by a function of randomly sampled k data points for each class. What if these samples do not provide a good approximation of the entire class?

3. Are unlabeled samples used in the pairwise representation learning stage? If yes, and pseudo-labels are used, there's a possibility that these labels are not accurate, affecting the accuracy of data point representations.

4. The study seems to lack other ablation studies. For example, what if larger encoder models are used, especially considering that results are only marginally improved in some cases?

5. Additionally, a suggestion: The author extensively uses bulk citations, such as at line 103. It might be beneficial to provide a brief one-liner about each paper for clarity.

**Reviewer Confidence:**

3: The reviewer is confident but not certain that the evaluation is correct

**Scope:**

3: The work is somewhat relevant to the Web and to the track, and is of narrow interest to a sub-community

---

### Official Review · Reviewer_Wmgo · 2023-12-01

**Novelty:** 5
**Technical Quality:** 5

**Review:**

### Overview and claims of the work

The paper aims to address some shortcomings in the area of semi-supervised text classification (SSTC) by aiming to reduce data annotation costs via a simple parameter free and adaptive approach. Existing approaches in the domain of SSTC train a classifier on limited labeled data and utilize it to generate labels which render them prone to overfitting. This work aims to address those issues by learning an adaptive prototype/exemplar based classifier via affinity learning. To do so they propose a Self-Paced Pair Wise (SPPW) representation learning model which utilizes a pairwise learning objective on representation learning. SPPWs combination of prototype calibration, confidence based filtering and confidence paced learning helps it achieve good performance of SSTC tasks on 3 datasets under varying settings of semi-supervision w.r.t. baselines.


### A list of Pros and Cons of the work

### Pros

* The authors approach of prototype based semi supervision helps avoid overfitting issues accompanying parameteric classifiers in scarce label domains.

* The design of parameter free classifier is simple, well motivated and grounded in terms of representation similarity/affinity learning and can be achieved without added signicant parameter overhead to baseline encoder models.

* Prototype based label inference and self paced training and filtering procedure are properly motivated, though the writing in section 4.1 and 4.2 could be improved a little to better connect the 2 stage training procedure and various loss functions.

* Despite the model's reliance on various hyperparameters in the training objective to achieve its desired performance, they are all well motivated with careful ablation studies which justify their inclusion and other experiments which explain their working.

### Cons

* The spectrum of baselines chosen for comparison seems less especially given that base model BERT could also be replaced by variants like ALBERT[1], DistilBERT[2], ELECTRA[3], RoBERTa[4].

* The targets for baselines for SSTC could be improved for instance with the addition of [5,6].

* The problem setup SSTC seems ripe for the application of Incontext/Few shot learning based methods though Language Models. It would indeed be valuable to do such a comparison to strengthen and better position the paper and is missing in the paper.




### References

[1] Zhenzhong Lan, Mingda Chen, Sebastian Goodman, Kevin Gimpel, Piyush Sharma, & Radu Soricut. (2020). ALBERT: A Lite BERT for Self-supervised Learning of Language Representations.

[2] Victor Sanh, Lysandre Debut, Julien Chaumond, & Thomas Wolf. (2020). DistilBERT, a distilled version of BERT: smaller, faster, cheaper and lighter.


[3] Kevin Clark, Minh-Thang Luong, Quoc V. Le, & Christopher D. Manning. (2020). ELECTRA: Pre-training Text Encoders as Discriminators Rather Than Generators.

[4]Yinhan Liu, Myle Ott, Naman Goyal, Jingfei Du, Mandar Joshi, Danqi Chen, Omer Levy, Mike Lewis, Luke Zettlemoyer, & Veselin Stoyanov. (2019). RoBERTa: A Robustly Optimized BERT Pretraining Approach.


[5] Weifeng Jiang, Qianren Mao, Chenghua Lin, Jianxin Li, Ting Deng, Weiyi Yang, & Zheng Wang. (2023). DisCo: Distilled Student Models Co-training for Semi-supervised Text Mining.

[6] Pavel Izmailov, Polina Kirichenko, Marc Finzi, & Andrew Gordon Wilson. (2019). Semi-Supervised Learning with Normalizing Flows.

**Questions:**

### Questions and comments

Before commencing I would urge the authors to reconsider the presentation and writing especially around Sec 4.1 and 4.2 in terms of the model training pipeline. If possible, the authors can also utilize the space in the appendix to make the training process a bit more clear and use it as a space to collect and reorganize the flow of information from Eq.1 till Eq. 12.


In Eq. 3 of the paper, the authors optimize for the inner product between vectors based on the desired similarity. Is this choice motivated by any particular advantages? Say over other choices like the RBF Kernel or cosine similarity etc.?

Regarding my comments about baselines it would be helpful to understand the choices made by the authors in selecting them for a better contextualization of my understanding.

**Reviewer Confidence:**

3: The reviewer is confident but not certain that the evaluation is correct

**Scope:**

3: The work is somewhat relevant to the Web and to the track, and is of narrow interest to a sub-community

---

### Official Review · Reviewer_nqjd · 2023-12-02

**Novelty:** 4
**Technical Quality:** 5

**Review:**

This paper presents a new Semi-supervised Text Classification (SSTC) method that trains classifiers on limited labeled data supplemented by pseudo-labeled data from unlabeled texts. The authors propose a Self-Paced PairWise (SPPW) model to address the “overfitted classifier” issue by focusing on representation learning rather than parameterized classifier learning. One key contribution of this work is a novel text filtering method for identifying “high-quality” pseudo-labeled examples. Specifically, it combines label confidence and text hardness to minimize mislabeling. The effectiveness of SPPW is demonstrated through superior performance over baseline methods in experiments on multiple benchmark SSTC datasets.

Overall the paper is clearly written and easy to digest. The overall framework is somewhat standard but reasonable. The experiments are overall well designed, although it’s better to test model performance on more large-scale backbone models (at least to BERT-large sized model, better with t5 xxl size or llama 2 small size).

My two comments on the paper weakness side are: (1) the claim of “overfitted classifier” issue which leads to the adoption of prototype learning based framework is not very convincing. Typically with the BERT-based text encoder, the final “classifier” is just a linear model which has a very limited number of parameters. When given a small labeled example set, the overfitting problem could happen, but more likely for the entire encoder (if that module is fine-tuned) not just the classifier. Furthermore, I don’t see why the data filter techniques introduced in the other part of paper cannot be applied to a traditional parameterized classifier and have to be combined with the current non-parametric prototypical network. (2) The key novelty on the “Self-Paced” seems a little bit weak. I feel it’s more like a new data filtering rule (eq. 12) added on top of the previous standard confidence-based rule. More explanations on this novelty can help boost the paper's significance.

**Questions:**

Have you tried your method on more large scale models and/or LLMs?
Is the “confidence-aware value adaptation strategy” introduced around line 473-480 very important? What happens if you remove this part?

**Reviewer Confidence:**

3: The reviewer is confident but not certain that the evaluation is correct

**Scope:**

3: The work is somewhat relevant to the Web and to the track, and is of narrow interest to a sub-community

---

### Decision · Program_Chairs · 2024-01-22

**Decision:**

Accept

**Comment:**

The paper presents the Self-Paced Pairwise Representation Learning (SPPW) model for semi-supervised text classification. The SPPW model addresses two key issues in this field: overfitting due to limited labeled data and the mislabeling problem stemming from unreliable pseudo-labeling. The approach is innovative in using pairwise representation learning to reduce overfitting and introducing a novel self-paced text filtering method that integrates label confidence and text hardness. The paper's experiments demonstrate the model's effectiveness, significantly outperforming baselines on benchmark datasets.

 Strengths:
 1. Innovative approach to address overfitting and mislabeling in semi-supervised text classification.
 2. Comprehensive experimental validation showing effectiveness against baseline models.


 The reviewers had some concerns regarding the paper:

 1. **Overfitted Classifier Issue**: Some reviewers were not convinced that the classifier is the main component prone to overfitting, suggesting that the entire encoder might be affected. They questioned the necessity of replacing a traditional parameterized classifier with the proposed model.

 2. **Novelty of Self-Paced Learning**: Reviewers felt that the novelty claimed in the self-paced learning component was weak, resembling more a data filtering rule rather than a novel approach.


 3. **Writing and Presentation**: Clarity in writing, especially in sections detailing the model training pipeline, was suggested for improvement.


 Overall, I think this paper makes a nice contribution and the authors responded well to the reviewer concerns.